# Data Augmentation for Code Translation with Comparable Corpora and Multiple References

**Yiqing Xie** **Atharva Naik** **Daniel Fried** **Carolyn Rosé**
Language Technologies Institute
Carnegie Mellon University
{yiqingxi, arnaik, dfried, cprose}@cs.cmu.edu

## Abstract

One major challenge of translating code between programming languages is that parallel training data is often limited. To overcome this challenge, we present two data augmentation techniques, one that builds comparable corpora (i.e., code pairs with similar functionality), and another that augments existing parallel data with multiple reference translations. Specifically, we build and analyze multiple types of comparable corpora, including programs generated from natural language documentation using a code generation model. Furthermore, to reduce overfitting to a single reference translation, we automatically generate additional translation references for available parallel data and filter the translations by unit tests, which increases variation in target translations. Experiments show that our data augmentation techniques significantly improve CodeT5 for translation between Java, Python, and C++ by an average of 7.5% Computational Accuracy (CA@1), which verifies the correctness of translations by execution.[1]

## 1 Introduction

Code translation is a special type of machine translation that translates between programming languages. It is widely applied in software engineering to migrate a codebase into another programming language. Recent code translation models typically follow the pretrain-finetune pipeline, as shown in Figure 1. In pretraining, with denoising objectives such as masked span or identifier prediction (Ahmad et al., 2021a; Wang et al., 2021; Roziere et al., 2020), the model learns to produce sequences in both languages. When finetuned on parallel data (Ahmad et al., 2021b), which are program pairs in the source and target language that are aligned line-by-line, the model learns functional equivalence: identifying programs with the same functionality,

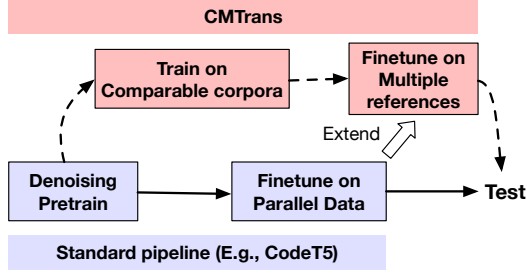

Figure 1: The standard pipeline for code translation and the pipeline of CMTrans. The comparable corpora are both naturally occurring and model generated. We generate multiple references by our method.

either in the same language or between languages. We show an example of parallel data in Figure 2.

One major challenge of code translation is that parallel data is typically limited. For instance, the TransCoder dataset (Roziere et al., 2020) only contains 466 Python-C++ pairs. Constructing parallel data requires substantial human effort and cannot be easily scaled. With limited fine-tuning examples, it is difficult for a model to learn functional equivalence across programming styles and domains.

To mitigate the data scarcity issue, we hypothesize that the functional equivalence between languages can also be learned from *comparable corpora*, a term we borrow from natural language translation, where it refers to texts on similar topics in different languages (Gete and Etchegoyhen, 2022; Irvine, 2014). Here, we use it to refer to programs with similar functionality in different languages (Ahmad et al., 2021b; Khan et al., 2023). As shown in Figure 2, although the programs paired in a comparable example may have different algorithms and structures (e.g., global code vs. class functions), they are likely to have similar constructs and may even have some lines that can be matched.

In this paper, we *study what the model learns from comparable corpora* by building three types of comparable examples: (1) *Naturally existing*, where we leverage independently-written solutions

---

[1] Code available at https://github.com/Veronicium/CMTrans.

```
board = []                  Source (Python)
check = False

for i in range (8):
    board.append( str(input()) )
for x in board:

    if "BB" in x or "WW" in x:
        print ( "NO" )
        check = True
        break

    if x != "BW"*4:
        if x != "WB"*4:
            print( "NO" )
            check = True
            break

if check == False:
    print( "YES" )
```

```
String[] board = new String[8];   Parallel (Java)
Boolean check = false;
Scanner in = new Scanner( System.in );
for( int i = 0; i < 8; i++ )
    board[i] = in.next();
for (int i = 0; i < 8; i++){
    String x = board[i];
    if (x.contains("BB") || x.contains("WW")){
        System.out.println( "NO" );
        check = true;
        break;
    }
    if (!x.equals("BW".repeat(4)))
        if (!x.equals("WB".repeat(4))){
            System.out.println( "NO" );
            check = true;
            break;
        }
}
if (check == false)
    System.out.println( "YES" );
```

```
import java.util.*;              Comparable (Java)
import java.lang.*;
import java.io.*;
public class Ideone {
    public static void main( String[] args ){
        Scanner in = new Scanner( System.in );
        int f = 0;
        for( int i = 0; i < 8; i++ ){
            String s = in.next();
            in.nextLine();
            for ( int j = 0; j < 7; j++ ){
                if ( s.charAt(j) == s.charAt(j+1) ){
                    f++;
                }
            }
        }
        if(f == 0) System.out.println( "YES" );
        else System.out.println( "NO" );
    }
}
```

Figure 2: An example of parallel and comparable data. Parallel examples are line-by-line aligned. Programs in a comparable example may have different algorithms and structures (e.g., global code vs. class in this case), but may still contain lines that can be matched, as highlighted in yellow, green, and blue.

of the same coding problem; (2) *Generated*, where we collect programs with docstrings in one language and apply a code generation model to generate programs in another language; and (3) *Retrieved*, where we either retrieve a program's k nearest neighbors (KNN) or simply choose a random program in another language. Among them, (1) contains cleaner examples, which are guaranteed to be bug-free. (3) covers programs from a larger variety of sources, providing more diverse training signals.

In addition to the functional equivalence between languages, the model should also learn the equivalence between different programs in the target language. This is challenging due to limited finetuning data. Furthermore, the majority of finetuning data only have one reference translation (e.g., 82.5% in AVATAR, Ahmad et al. 2021b), which is likely to cause overfitting to a single translation without fully capturing its semantics.

As a result, in this paper, we *generate multiple translation references* for the finetuning data. Specifically, after training on comparable corpora, we finetune a model on the original parallel data, generate multiple translations for each example, and use automatically generated test cases to filter out incorrect translations. By training on different functionally equivalent programs, we reduce overfitting to a single output and improve the modeling of the target language.

Combining the two techniques, we name our full approach CMTrans, a code **Trans**lation model trained with **C**omparable corpora and **M**ultiple references. Extensive experiments show that CM-Trans significantly improves CodeT5 (Wang et al.,

2021), which is initialized from the same pretrained checkpoint, for an average of 7.5% Computational Accuracy (CA@1) over translation between 6 language pairs. CMTrans also significantly outperforms the state-of-the-art method in 5 out of 6 language pairs, while reaching parity on the other one.

Analyses of our two techniques suggest that: (1) All three types of comparable corpora (including random program pairs) improve the syntax accuracy and perplexity of the translation outputs and lead to better final performance. (2) Both naturally existing and generated comparable corpora help the model generate constructs that match the input. The combination of them gives the largest performance gain. (3) By training with multiple references, the model generates more unique correct translations within a certain budget, which indicates better functional equivalence of the target language is learned.

**Contributions**. (1) We build and study three types of comparable corpora for code translation. (2) We improve the modeling of target language by generating, verifying, and selecting additional references for existing parallel translation data. (3) We demonstrate that our model significantly outperforms state-of-the-art methods on 5 language pairs. Our analyses provide insights into what the model learns in code translation for future researchers.

## 2 Related Work

**Code Translation**. Previous work has tackled the problem of translating code written in one programming language to another. Karaivanov et al. (2014), Nguyen et al. (2013, 2015), Phan et al. (2017); Oda

et al. (2015) applied statistical machine translation techniques to code translation, while Chen et al. (2018) introduced a tree-to-tree neural translation approach. Further improvements were achieved by pre-trained language models of code such as Code-BERT (Feng et al., 2020), PLBART (Ahmad et al., 2021a), and CodeT5 (Wang et al., 2021). However, the above approaches require finetuning on parallel data, which is often scarce.

**Data Scarcity in Code Translation**. To tackle the data scarcity issue, TransCoder (Roziere et al., 2020) uses back translation for unsupervised code translation. DOBF (Lachaux et al., 2021), TransCoder-ST (Roziere et al., 2022), and S&G (Ahmad et al., 2022) respectively improve TransCoder with de-obfuscation pre-training, self-training, and pairing up the model with a generation and a summarization model. However, the best-performing approach, TransCoder-ST, is only able to generate parallel data for standalone functions where the model can already generate a correct solution within a limited budget. In contrast, the comparable corpora we use to train CMTrans include code with arbitrary structure and content. CMTrans also has much better efficiency, as self-training requires running a much larger number of test cases than generating multiple references. MultiPL-E (Cassano et al., 2022) also automatically generates test cases, but retries until the output passes all test cases. We do not directly compare to it since it requires multiple rounds of translation.

**Data Augmentation for Natural Language Translation**. The lack of parallel training data is a fundamental problem in the field of translation, which has led the NLP community to develop several data augmentation techniques in response. *Comparable Corpora*, as defined by Munteanu and Marcu (2005), "are texts that, while not parallel in the strict sense, are somewhat related and convey overlapping information". Paramita et al. (2013) and Wołk et al. (2015) present methods for collecting comparable corpora to study when and how to use them. Etchegoyhen and Gete (2020) and Gete and Etchegoyhen (2022) identify information balance, alignment, and length differences between source and target as key factors affecting translation quality. In this work, we extend the study of comparable corpora to code translation. We show that code translation can benefit from multiple types of comparable corpora even if there is already high-quality parallel data. We also provide analyses on

why comparable corpora are beneficial.

Another data augmentation strategy is using multiple translation references. Qin and Specia (2015) and Khayrallah et al. (2020) found that using multiple references can be beneficial in low-resource settings. An advantage of working with code is that test cases can be used to filter model-generated translated references to ensure functional equivalence to the source, which we exploit in CMTrans.

## 3 Methodology

In this section, we introduce our data augmentation method that trains the model first on comparable corpora, which are either naturally existing, generated, or retrieved (Section 3.2) and then on parallel data with additional references generated by our model (Section 3.3).

### 3.1 Problem Formulation

We formulate code translation as a sequence-to-sequence (Seq2Seq) generation problem following existing work (Lu et al., 2021; Ahmad et al., 2021b; Roziere et al., 2020). The input is the code in the source language $S = \{s_1, s_2, ..., s_n\}$, which is a sequence of tokens. We apply a model to translate the input code into the target language $T = \mathcal{M}_\theta(S) = \{t_1, t_2, ..., t_m\}$, where the model $\mathcal{M}_\theta$ is typically finetuned for code translation. Alternatively, the model may generate $k$ candidate outputs for each input: $\mathcal{M}_\theta(S, k) = \mathbb{T} = \{T^{(1)}, T^{(2)}, ..., T^{(k)}\}$.

### 3.2 Training on Comparable Corpora

As shown in Figure 3, we study three types of comparable corpora: naturally existing, generated, and retrieved ones. Here we introduce how we build each type of comparable examples and train our model on them. We provide details of the comparable corpora datasets we use or build in Section 4.1 and provide examples in Appendix A.3.

**Naturally Existing Comparable Examples**. We make use of comparable corpora collected from programming contests by existing datasets (Ahmad et al., 2021b; Khan et al., 2023; Puri et al., 2021), which mainly consist of solutions to the same contest problems written in different languages. These comparable examples are typically confined to specific domains such as dynamic programming or graph theory.

**Generated Comparable Examples**. To cover programs in more diverse domains, we present a method that automatically generates comparable

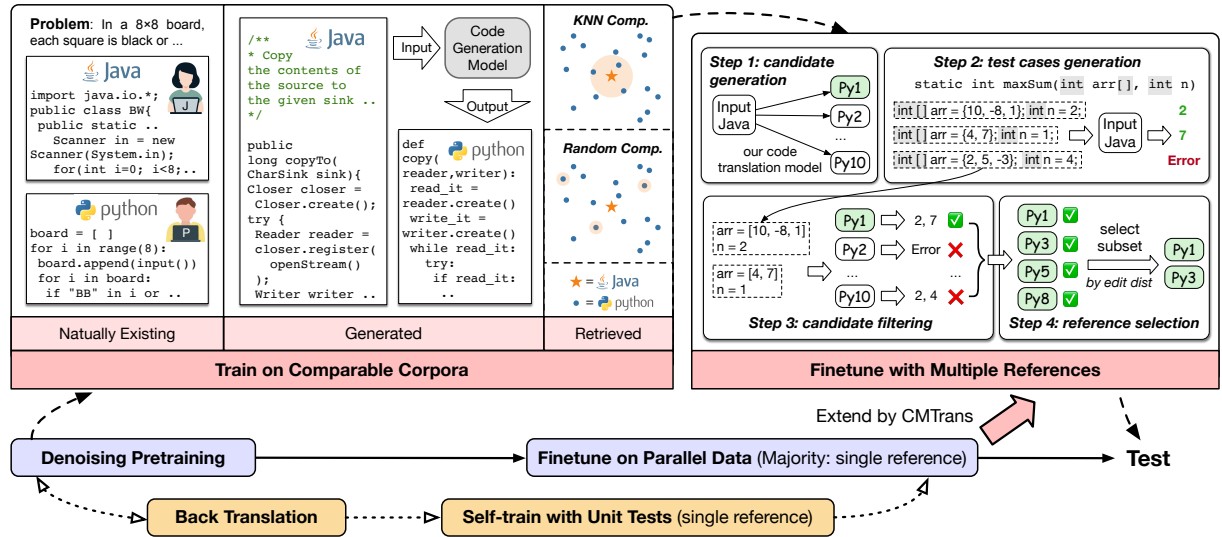

Figure 3: An example of CMTrans for Java-to-Python translation. We compare the pipeline of CMTrans to the standard pipeline of code translation (e.g., CodeT5, Wang et al. 2021) and the self-supervision-and-fine-tuning method of TransCoder-ST (Roziere et al. 2022).

examples using natural-language-to-code generation models.

Specifically, we leverage a monolingual corpus of functions with natural language documentation (i.e., docstrings) to describe their functionality. In our experiments, we use GitHub functions with docstrings extracted by CodeSearchNet (Husain et al., 2020). For each function $S_c$ in the corpus, we feed the natural language documentation to a code generation model finetuned in the other language, which is trained to generate a program based on the given natural language description. Similarly to code translation, the code generation model can generate multiple candidate outputs $\{T_c^{(1)}, T_c^{(2)}, ..., T_c^{(k)}\}$. We then select the one with the highest probability, which is paired up with the extracted program as a comparable example.

**Retrieved Comparable Examples**. To study the effect of the quality of comparable corpora, we further build **KNN Comparable Corpora**. We compute the embedding of all the programs in the source and target language in the dataset using a finetuned code retrieval model (Husain et al., 2020). For each program $S$ in the source language with embedding $emb(S)$, we retrieve its k nearest neighbors in the target language by the cosine similarity between their embeddings: $sim(S, T) = \langle emb(S), emb(T) \rangle$.

Finally, we build **Random Comparable Corpora** by pairing random programs in the source and target language. In principle, such program pairs

do not contain any information on functional equivalence and allow us to better understand whether comparable corpora can improve the modeling of the target language and hence improve the translation quality.

**Model Training**. After constructing a corpus of comparable examples $\mathcal{D}_c = \{(S_c, T_c)\}$, we input the program in the source language $S_c$ into our model $\mathcal{M}_\theta$ and maximize the log-likelihood of its corresponding program in the target language $T_c = \{t_{c,1}, t_{c,2}, ..., t_{c,m}\}$:

$$\mathcal{L}_{MT}(\mathcal{D}_c) = \sum_{(S_c, T_c) \in \mathcal{D}_c} P_\theta(T_c | S_c)$$

$$P_\theta(T_c | S_c) = -\sum_i \log\left(P_\theta\left(t_{c,i} | S_c, t_{c,1} \ldots t_{c,i-1}\right)\right)$$

(1)

Here $\theta$ is the parameters of $\mathcal{M}_\theta$, which is initialized from an encoder-decoder model pretrained on code (Wang et al., 2021). After training the model on the corpus comparable examples $\mathcal{D}_c$ till convergence, we finetune it on the dataset of parallel examples $\mathcal{D}_p$, where the loss $\mathcal{L}_{MT}(\mathcal{D}_p)$ is computed by Equation 1 as well.

By maximizing the probability of the target $T_c$, the model will learn to generate fluent code in the target language. Furthermore, in general, programs in a comparable example often exhibit the same types of constructs. In the examples in Figure 2, to check whether a board is valid, no matter what algorithm is used, the program will always need

| Dataset → | Train (Comparable) | | | Train (Parallel) | Test |
|---|---|---|---|---|---|
| | Gen-comp | XCodeEval | AVATAR-comp | AVATAR-para | TransCoder-test |
| # C++ ↔ Java | 4,053* | 3,414 | – | 3,226* | 482 C-to-J, 467 J-to-C |
| # C++ ↔ Python | 4,053* | 4,376 | – | 3,226* | 464 C-to-P, 467 P-to-C |
| # Java ↔ Python | 22,181* | – | 5,937 | 3,391 | 464 J-to-P, 482 P-to-J |
| Source | Github (Java ↔ Python) AIZU, AtCoder (Others) | Codeforces | AtCoder, Codeforces, ProjectEuler, CodeJam, GeeksforGeeks, LeetCode, | GeeksforGeeks | GeeksforGeeks |

Table 1: Number of problems per dataset. "Gen-comp" is the comparable corpora dataset we generate. C-to-J, C-to-P, ... denote the test examples of C++-to-Java, C++-to-Python translation, etc. * denotes data we build.

a loop to read the board and use if statements for the validity check. As a result, in principle, the model can also learn to generate the same types of constructs as the input program $S_c$, which is beneficial for generating accurate translations.

### 3.3 Finetuning with Multiple References

In addition to comparable corpora, we also provide our model with more diverse training signals by finetuning with multiple references. By providing the model with programs with the same functionality, we encourage the model to learn a better representation space for the target language and hence benefit the translation.

Since the majority of source programs only have one reference in existing datasets of parallel examples (Ahmad et al., 2021b), we apply a series of steps to generate additional references, which are illustrated in Figure 3. The first step is to finetune a model with the original parallel data, and then use the finetuned model to generate multiple translation candidates $\{T_c^{(1)}, T_c^{(2)}, ..., T_c^{(k)}\}$ for each source program in the parallel data.

In the second and third steps, similar to TransCoder-ST (Roziere et al., 2022), we leverage automatically generated unit tests to filter out candidates with different behaviors as the source program. Specifically, we extract the input arguments of the source programs, randomly produce a set of test inputs based on the argument types, and feed the test inputs to the source program. We filter out test inputs that cause compilation or runtime errors. Finally, we feed the remaining test inputs to each candidate and only keep the ones that have exactly the same output as the source program.

Notice that some of the translations may only have small differences (e.g., `i++` vs. `i+=1`). To obtain a diverse subset of references, we select the most distinct $k$ translations for each source program by their string edit distance. These $k$ translations are added as additional references to the finetuning set.

## 4 Experiments

In this section, we conduct experiments to answer four research questions: (**RQ1**) How do CMTrans and its ablations perform compared against state-of-the-art approaches for code translation? (**RQ2**) What can the model learn from comparable corpora? (**RQ3**) What can the model learn from multiple references? (**RQ4**) How is CMTrans affected by the size of comparable corpora and the number of references?

### 4.1 Experimental Setup

We initialize CMTrans from the pretrained checkpoint of CodeT5 (Wang et al., 2021), an encoder-decoder model pretrained on code files from varied languages with denoising objectives.

**Datasets**. We list the dataset statistics in Table 1. All the methods are evaluated on the TransCoder-test dataset (Roziere et al., 2020). We train our method on xCodeEval (Khan et al., 2023), a comparable corpora dataset, and AVATAR (Ahmad et al., 2021b), which contain both comparable corpora and parallel functions (denoted as AVATAR-comp and AVATAR-para, respectively). All the supervised baseline methods are finetuned on AVATAR-para before evaluation.

Since AVATAR-para does not contain C++ functions, we add a parallel C++ function for each training example in AVATAR-para. Specifically, we generate 50 C++ translations for each Java function by TransCoder-ST, TransCoder, and finetuned CodeT5 and filter the translations with unit tests.

**Construction of Comparable Corpora**. We conduct the study on different types of comparable corpora (Comp-Corp) on Java ↔ Python translation. We use AVATAR-comp[2] as the naturally existing Comp-Corp. KNN and random Comp-Corp are also retrieved from AVATAR-comp. To build the

---
[2]There are two versions of AVATAR-comp with slight differences. We use the first version because most of our experiments were finished before the second version was released.

| Model ↓ | Java-to-Python | | | Python-to-Java | | | Avg of 6 Pairs | | |
|---|---|---|---|---|---|---|---|---|---|
| | BLEU | CB | CA@1 | BLEU | CB | CA@1 | BLEU | CB | CA@1 |
| TransCoder (Roziere et al., 2020) | 72.4 | 67.9 | 49.1 | 65.4 | 70.7 | 35.7 | 72.0 | 75.0 | 51.7 |
| DOBF (Lachaux et al., 2021) | 72.2 | 67.5 | 52.2 | 67.7 | 71.2 | 44.4 | — | — | — |
| TransCoder-ST (Roziere et al., 2022) | 73.1 | 68.7 | 68.5 | 70.0 | 71.9 | 58.1 | 71.3 | 74.9 | 66.3 |
| CodeBERT (Feng et al., 2020) | 52.0 | 48.9 | 10.4 | 45.4 | 45.0 | 4.2 | — | — | — |
| CodeT5 (Wang et al., 2021) | 79.4 | 72.5 | 61.0 | 79.0 | 75.9 | 52.7 | 83.6 | 80.0 | 62.6 |
| PLBART (Ahmad et al., 2021a) | 79.9 | 73.2 | 68.9 | 80.5 | 76.8 | 57.5 | — | — | — |
| TransCoder-ST-ft (Roziere et al., 2022) | 79.3 | 72.9 | 69.4 | 81.4 | 78.4 | 62.0 | 81.8 | 80.2 | 67.6 |
| CMTrans | **80.1** | **74.2** | **73.5** | **84.3** | **82.1** | **66.0** | **84.9** | **82.0** | **70.1** |

Table 2: Java-Python translation results on TransCoder-test. We copy the results of all the baselines reported by Ahmad et al. (2021b). CB and CA@1 stand for CodeBLEU and Computational Accuracy. We highlight the **best** results under each metric with Bold and underline the second-best results.

| Model ↓ | C++-to-Java | | | C++-to-Python | | | Java-to-C++ | | | Python-to-C++ | | |
|---|---|---|---|---|---|---|---|---|---|---|---|---|
| | BLEU | CB | CA@1 | BLEU | CB | CA@1 | BLEU | CB | CA@1 | BLEU | CB | CA@1 |
| TransCoder (Roziere et al., 2020) | 84.0 | 86.7 | 65.1 | 75.2 | 73.4 | 47.1 | 83.6 | 85.4 | 79.8 | 51.6 | 65.7 | 32.6 |
| TransCoder-ST (Roziere et al., 2022) | 78.8 | 85.2 | 68.0 | 73.1 | 73.0 | 61.3 | 76.7 | 83.7 | **84.6** | 55.8 | 67.1 | 56.7 |
| CodeT5 (Wang et al., 2021) | 90.9 | 90.0 | 65.1 | 82.9 | 75.4 | 56.5 | 89.1 | **88.5** | 81.6 | 79.8 | 77.9 | 58.5 |
| TransCoder-ST-ft (Roziere et al., 2022) | 88.7 | 90.1 | 68.3 | 75.4 | 74.3 | 62.5 | 89.3 | 87.8 | **84.6** | 76.7 | 77.6 | 59.1 |
| CMTrans | **91.6** | **90.5** | **71.4** | **83.7** | **77.2** | **64.2** | **89.7** | 88.4 | 84.4 | **82.0** | **79.3** | **61.2** |

Table 3: Java-C++ and Python-C++ translation results on TransCoder-test. The CA@1 results of TransCoder and TransCoder-ST are copied from the TransCoder-ST paper. We evaluate BLEU and CodeBLEU using their released checkpoints. We finetune and evaluate CodeT5 and TransCoder-ST-ft on our own.

dataset of generated Comp-Corp (denoted as Gen-Comp), we generate from functions with docstrings in the CodeSearchNet (Husain et al., 2020) dataset.

We train CMTrans first on the best combination of comparable corpora, which is natural and generated Comp-Corp, and then finetuned on AVATAR-para. For language pairs other than Java ↔ Python, we use xCodeEval as naturally existing Comp-Corp and generate Comp-Corp from CodeNet (Puri et al., 2021). More details can be found in Appendix A.1. **Evaluation metrics**. Our primary metric is the Computational Accuracy (CA@k), which evaluates whether at least 1 out of k translation candidates generates the same outputs as the reference for all the given test inputs. Following previous work (Ahmad et al., 2021b), we report **CA@1** results and also report **BLEU**, which computes the overlap between candidate and reference translations (Papineni et al., 2002), and **CodeBLEU**: the weighted average of token level match, syntax level match, and Dataflow match (Ren et al., 2020).

Appendix A.1 contains more details about the implementation and baselines.

### 4.2 Main results

Table 2 and Table 3 show the code translation results on TransCoder-test.

CMTrans substantially outperforms CodeT5, which is initialized from the same pretrained checkpoint and finetuned with the original parallel data, by an average improvement of 7.5% CA@1. CMTrans also significantly outperforms the state-of-the-art methods, TransCoder-ST-ft, on 5 out of 6 language pairs, and reaches parity on Java-to-C++ translation. Note that TransCoder-ST-ft generates test cases for 103,488 functions and executes these test cases over 4 iterations of training. In comparison, our method only generates test cases for 3,391 functions and executes the test cases once, resulting in better efficiency.

Our results show that the advantage of CMTrans is larger on translations between Java and Python. The reason may be that the parallel data we generate for translations involving C++ are only verified by automatically generated test cases, which might not cover all the boundary cases and could introduce noise to the finetuning set.

Following previous work (Roziere et al., 2022; Ahmad et al., 2021b), we report the results of one checkpoint. We also conduct the t-test in Appendix A.2, which indicates that CMTrans significantly outperforms the best baseline over CA@1 with p-value $< 0.01$ for 4 out of 6 language pairs and with p-value $< 0.05$ for one language pair.

| Model ↓ | J-to-P CA@1 | P-to-J CA@1 |
|---|---|---|
| CodeT5 | 61.0 | 52.0 |
| + Random Comp-Corp | 64.7 | 54.6 |
| + KNN Comp-Corp | 63.8 | 55.4 |
| + Generated Comp-Corp | 64.0 | 63.1 |
| + Natural Comp-Corp | 68.8 | 62.9 |
| + All Comp-Corp | 62.5 | 55.4 |
| + Natural & Generated | 70.0 | 64.1 |
| + Multi-Ref | 68.1 | 61.6 |
| CMTrans | **73.5** | **66.0** |

Table 4: Ablation studies of CMTrans. We report CA@1 results. J-to-P and P-to-J stand for Java-to-Python and Python-to-Java results. We put the experiments on each type of comparable corpora in an individual block.

## 4.3 Performance Analysis

**Ablation studies**. Table 4 presents the ablations of our approach. We first study the effectiveness of each type of comparable corpora and some combinations of them. All the "+ Comp-Corp" ablations denote training CodeT5 on comparable corpora and then finetuneing on AVATAR-para. We also ablate "+ Multi-Ref", which is finetuned on multiple references directly after pretraining. The results show that both Multi-Ref and the best Comp-Corp provide a large performance gain and stacking them (the full CMTrans) has the best performance.

As for the effectiveness of different Comp-Corp, we can see that code translation can benefit from all these types of Comp-Corp. The combination of Natural and generated Comp-Corp has the best performance. The reason might be both Comp-Corp have relatively high data quality and further adding the other two Comp-Corp in training introduces noise to the model and hinders the learning of functional equivalence.

Notice that with the same finetuning set, *Comp-Corp Only* still outperforms TransCoder-ST-ft, which indicates that compared to self-training, training on comparable corpora has not only better efficiency but better effectiveness. We hypothesize that the reason may be that the comparable corpora contain more diverse structures and contents. This provides more diverse training signals to the model and hence improves the generalization.

**Translation with Limited Parallel Data**. To analyze how well our method tackles the data scarcity challenge, we compare the performance of CodeT5 and CMTrans when finetuned on parallel data with different sizes. For simplicity, we use CMTrans (Comp-Corp Only) to denote the "CodeT5 + Natu-

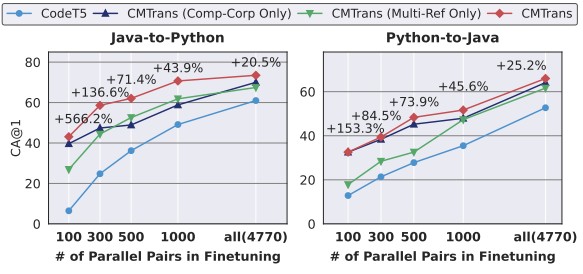

Figure 4: Translation results with different amount of parallel data. We mark the relative gain of CMTrans over CodeT5.

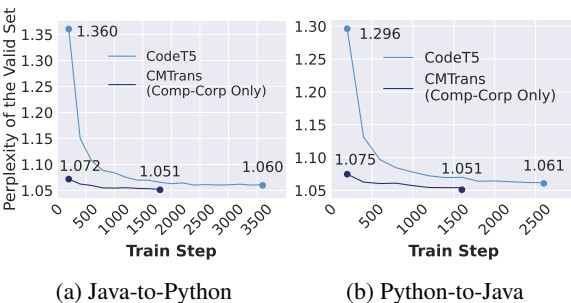

(a) Java-to-Python      (b) Python-to-Java

Figure 5: Perplexity of validation set during finetuning.

ral & Generated" ablation and use CMTrans (Multi-Ref Only) to denote "CodeT5 + Multi-Ref".

As shown in Figure 4, the relative gains of CM-Trans as well as our ablations are more pronounced when the parallel data is more limited. For example, when there are only 100 parallel examples for Java-to-Python translation, CodeT5 obtains 6.5 CA@1 while CMTrans, (Comp-Corp Only), and (Multi-Ref Only) obtain 43.1, 39.7, and 26.9 CA@1, respectively. This demonstrates the effectiveness of our two data augmentation techniques to tackle data scarcity.

## 4.4 Influence of Comparable Corpora

To answer **RQ2**, we hypothesize that training on comparable corpora allows the model to produce fluent code in the target language while using similar constructs as the input.

**Fluency of outputs**. To validate our hypothesis, we compare the perplexity of the reference translations (in the target language) during finetuning, which reflects the fluency of outputs. As shown in Figure 5, after training on comparable corpora, the perplexity is substantially reduced before finetuning. It also converges to a lower value after finetuning on parallel data.

Similarly, Table 6 shows that in most scenarios, training on comparable corpora leads to fewer syn-

| Model ↓ | Java-to-Python | | | | | | | | | Python-to-Java | | | | | | | | |
|---|---|---|---|---|---|---|---|---|---|---|---|---|---|---|---|---|---|---|
| | LOOP | | | IF | | | ELSE IF | | | LOOP | | | IF | | | ELSE IF | | |
| | P | R | F1 | P | R | F1 | P | R | F1 | P | R | F1 | P | R | F1 | P | R | F1 |
| CodeT5 (No finetune) | 100.0 | 28.5 | 44.4 | 99.1 | 35.4 | 52.2 | 0.0 | 0.0 | 0.0 | 100.0 | 14.6 | 25.5 | 100.0 | 36.6 | 53.6 | 0.0 | 0.0 | 0.0 |
| + KNN Comp-Corp | 89.4 | 94.6 | 91.9 | 82.1 | 64.3 | 72.1 | 41.6 | 57.6 | 48.3 | 87.9 | 98.9 | 93.1 | 90.2 | 50.9 | 65.1 | 30.5 | 16.4 | 21.3 |
| + Generated Comp-Corp | 99.9 | 98.9 | **99.4** | 99.0 | 93.2 | **96.0** | 84.6 | 79.2 | **81.8** | 98.3 | 96.7 | **97.5** | 98.7 | 94.8 | **96.7** | 83.3 | 86.4 | **84.8** |
| + Natural Comp-Corp | 95.8 | 85.4 | 90.3 | 91.2 | 73.9 | 81.6 | 50.0 | 49.6 | 49.8 | 87.4 | 98.9 | 92.8 | 88.2 | 80.4 | 84.1 | 57.8 | 33.6 | 42.5 |

Table 5: The overlap between the types of constructs in the translation outputs and the ground truth translations.

| Model ↓ | J-to-P SA | P-to-J SA |
|---|---|---|
| No finetuning | | |
| CodeT5 (Wang et al., 2021) | 1.1 | 1.7 |
| + Random Comp-Corp | 20.0 | 41.3 |
| + KNN Comp-Corp | 22.0 | **56.4** |
| + Generated Comp-Corp | **41.4** | 43.2 |
| + Natural Comp-Corp | 34.1 | 54.6 |
| With finetuning | | |
| CodeT5 (Wang et al., 2021) | 95.3 | 69.7 |
| + Random Comp-Corp | 96.3 | 69.7 |
| + KNN Comp-Corp | 96.3 | 71.0 |
| + Generated Comp-Corp | **97.6** | 71.2 |
| + Natural Comp-Corp | 97.4 | **76.4** |

Table 6: Syntax Accuracy (SA) on TransCoder-test before finetuning, which evaluates whether the program can be compiled without syntax errors.

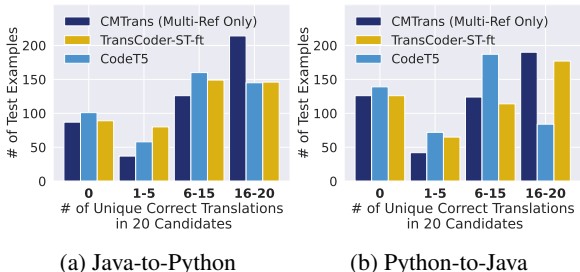

(a) Java-to-Python  (b) Python-to-Java

Figure 6: Number of unique correct translations in 20 candidates for each test example. We use beam search for each method, so the generated candidates are guaranteed to be distinct.

tax errors both before and after finetuning, which means the programs our method generates not only have a high token-level overlap with the reference translations but are also syntactically correct. The reason might be comparable corpora (including the random Comp-Corp) train the model to maximize the probability of a complete program in the target language, improving its language modeling ability.

**Generation of matching constructs**. We observe that programs in a comparable pair typically contain similar constructs (e.g., in AVATAR-comp, for 83.41% of Java programs with if statements, the corresponding Python programs also contain if statements). To assess whether our model also learns to generate the correct types of constructs, for each type of construct, we consider whether the reference translation has this type of construct as the ground truth and whether the translation output has it as the predictions. Then we compute the accuracy, recall, and F1 scores.

As shown in Table 5, after training on KNN, generated, and natural Comp-Corp, the F1 score is highly improved. The generated Comp-Corp has the highest F1 scores. The reason might be we input both the documentation and the program to

the generation model, so the generated program follows the same algorithm as the input program, which results in a larger percent of comparable examples that share the same types of constructs.

### 4.5 Influence of Multiple References

As for **RQ3**, we hypothesize that training on additional references can reduce overfitting to a single translation. To validate this hypothesis, we show in Figure 6 that when trained with multiple references, our model can generate more unique correct translations using beam search within the same number of candidate outputs. For instance, in Java-to-Python translation, there are 214 test examples where our model generates $\geq 16$ unique translations, while there are only 145 and 146 examples for CodeT5 and TransCoder-ST-ft, respectively. Furthermore, the scores under the "0" group indicate that after training with multiple references, the model generates at least one correct translation for more test examples. In other words, in addition to CA@1, training CodeT5 on multiple references also improves on CA@20.

### 4.6 Hyper-parameter analysis

To answer **RQ4**, we analyze two hyper-parameters: **Size of comparable corpora**. We sample the same percentage of comparable examples from AVATAR-comp and Gen-comp and combine the sampled

```
Returns a {@link Config} object from the supplied list with the supplied name, if it exists. If it exists, the supplied
list of {@code configs} will be modified such that it no longer contains the returned value.
                                                                                            <Input NL Documentation>
```

```java
public static Config getAndRemoveConfig(List <Config> configs, String name){
    final Config config = getConfigWithName(configs, name);
    if ( config != null ){ configs.remove( config ); }
    return config;
}
                                                                    <Input Java>
```

```python
def getAndRemoveConfig(configs, name):
    for i in configs :
        if i.name == name :
            configs.remove( i )
            return i
    return None                              <Generated Python>
```

Figure 7: Case study: a comparable example generated by our method with a NL-to-code generation model. We highlight the lines in the Java program and the generated Python program that can be matched.

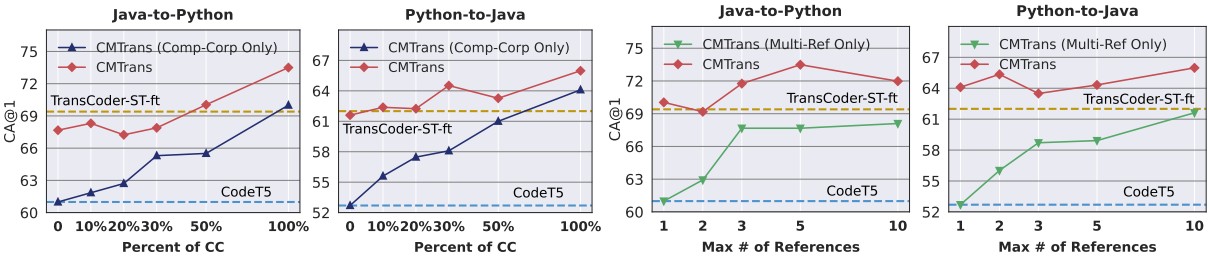

(a) Effect of the size of comparable corpora.

(b) Effect of the max number of references for each example.

Figure 8: The CA@1 score for CMTrans and its ablations with different hyper-parameters.

data. CMTrans is finetuned with at most 5 references in all trials. As shown in Figure 8a, as we increase the size of comparable corpora, CMTrans (Comp-Corp Only) always has better performance. While the trends are not monotonic for CMTrans, it still obtains the best performance with full comparable corpora.

**Number of references**. We also examine the effect of the maximum number of references for each parallel example in finetuning. We observe that the performance of CMTrans does not always increase when finetuned with more references. The reason might be our model may generate more unique correct translations for some training examples than others. As a result, the training signals the model received from different examples could be unbalanced, especially when the maximum number of references for each example is large.

### 4.7 Case Studies

We show a constructed comparable example in Figure 7. More case studies for comparable corpora and generated references can be found in Appendices A.3 and A.4. As shown in Figure 7, both the Java function extracted from GitHub and the generated Python function remove a config by name and return this config. The only difference is the Java function obtains the name of each config by a helper function, while the generated Python function directly accesses the "name" attribute. We also observe that this function is likely to belong

to a large software project, which has a different nature from coding problems (e.g., the example in Figure 2). As a result, combining collected and generated comparable examples provides diverse training signals to our model.

## 5  Conclusion and Future Work

We present two data augmentation techniques for code translation: comparable corpora and multiple references. We study multiple ways of building comparable corpora, such as feeding natural language code documentation to code generation models. Additionally, we generate multiple reference translations using our model and filter references using automatically generated unit tests. Experiments on translation between 6 language pairs show that our two techniques improve CodeT5 substantially, by an average of 7.5% CA@1. Our analyses further show that after training on comparable corpora, the model learns to generate more fluent code with the same types of constructs. With multiple references, the model learns to generate a larger number of unique correct translations.

Following our analyses of what the model learns from comparable corpora, future work may conduct a more comprehensive study on what the model learns in code translation. One may also explore combining our data augmentation techniques with data distillation from large language models (LLMs), where LLM may generate data with higher quality, but our techniques are less expensive.

## Limitations

Despite the empirical advantages of using comparable corpora (Comp-Corp) shown in our work, there are some inherent biases and limitations in how we collect and construct them. The collected Comp-Corp are from competitive programming websites, leading to a biased data distribution. The Comp-Corp constructed using code generation models are also biased by the training data seen by these models and can potentially contain errors. Furthermore, we only evaluate methods on the TransCoder dataset, which is currently the largest code translation dataset with test cases. The TransCoder dataset only contains standalone functions that don't contain any imports outside the standard libraries for each language. Translation of longer code with arbitrary external modules is an extension we plan to explore in future work. Another possible risk in our system is that the data may also contain information that uniquely identifies users or is offensive. For example, we generate comparable examples based on users' comments, which could contain inappropriate content.

## Ethics Statement

**License**. We use public datasets AVATAR (Ahmad et al., 2021b), xCodeEval (Khan et al., 2023), and TransCoder (Roziere et al., 2020) in our experiments. The data of AVATAR, xCodeEval, and TransCoder are all distributed under a CC BY-NC 4.0 license.

**Carbon Footprint**. We avoided the usage of large language models when constructing our models. CMTrans has the same architecture as CodeT5-base, which has 220 million parameters. The two models we use to construct comparable corpora have the same architecture as CodeT5-base and CodeT5-large, which have 220 million and 770 million parameters, respectively. We train CMTrans first on the comparable corpora and then the parallel data. Training on comparable corpora took 4-5 hours on average and training on parallel data took less than one hour on one RTX A6000 GPU. Therefore, training CMTrans results in approximately 0.78kg of carbon emission into the environment.[3]

---

[3]Estimations were conducted using the MachineLearning Impact calculator presented in Lacoste et al. (2019). We use Amazon Web Services as the provider.

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

## A Appendix

### A.1 Experimental Details

**Implementation details**. To generate the comparable corpora dataset (denoted as "Gen-comp" in Table 1), for Java $\leftrightarrow$ Python translation, we obtain Java functions with Docstrings from Code-SearchNet (Husain et al., 2020) and use CodeRL (Le et al., 2022) finetuned on MBPP (Austin et al., 2021) to generate Python programs. For Cpp $\leftrightarrow$ Java translation, we obtain Cpp solutions with problem descriptions from CodeNet (Puri et al., 2021) and use CodeT5 (Wang et al., 2021) finetuned on CONCODE (Iyer et al., 2018) to generate Java programs. For Cpp $\leftrightarrow$ Python translation, we again obtain Cpp programs from CodeNet and use the finetuned CodeRL model to generate Python programs.

To train CMTrans, we tune the number of source/target programs per problem in the range of [1, 3, 5] and tune the maximum number of generated references for finetuning in the range of [5, 10]. For the training on both comparable corpora and AVATAR-para, we train CMTrans with a learning rate of 1e-4 and batch size of 32 for at most 20 epochs. As for the baselines, we finetune CodeT5 with a learning rate of 1e-4 and batch size of 32 for at most 20 epochs. We finetune TransCoder-ST-ft with a learning rate of 1e-4 and batch size of 64 for at most 20 epochs.

**Baselines**. We compare CMTrans with unsupervised and self-supervised models, including TransCoder (Roziere et al., 2020), DOBF (Lachaux et al., 2021), and TransCoder-ST (Roziere et al., 2022). We also compare with supervised models, which are initialized from CodeBERT (Feng et al., 2020), PLBART (Ahmad et al., 2021a), CodeT5 (Wang et al., 2021), or TransCoder-ST (Roziere et al., 2022) (denoted as TransCoder-ST-ft) and finetuned on AVATAR-para. We only compare with DOBF, CodeBERT, and PLBART on Java $\leftrightarrow$ Python translation because these models are not pretrained on Cpp.

### A.2 Statistical Significance Test

We present the t-test results in Table 7, where we run each experiment with 3 random seeds when finetuning on AVATAR-para.

### A.3 Case Studies for Naturally Existing and Generated Comparable Corpora

We show case studies of naturally existing and generated comparable examples in Figure 9 and Figure 10. The example in Figure 9 is from the xCodeEval dataset, which contains different users' submissions of the "Food for Animals" problem on Codeforces. In Figure 10, the Java function and its docstring are from the CodeSearchNet dataset. The Python program is generated by our method.

**Quality of the Generated Comparable Example**. As shown in Figure 10, the Python program we generate has similar functionality as the input Java program. Specifically, both programs define a list of variables, compare the sum of term offset and aligned length with the term length, and return an offset of the term. The major differences are that the Java program calls a global function "getAndAddRawTail" to compute the value of "rawTail", while the generated Python program calls a class function "termBuffer.active()". Both functions are not defined in the context. Also, the Java program calls another function, "handleEndOfLogCondition", without defining it, while our method also generates the content of "handleEndOfLogCondition".

We notice that a large percentage of differences between the generated and input program are due to calling functions without presenting their definitions in the input. Such input programs contain limited information on the purpose of these functions. As a result, it is challenging for the code generation model to generate code with exactly the same functionality.

**Comparison Between Naturally Existing and Generated Comparable Examples**. We can observe that the naturally existing and generated comparable examples are different in several ways. For instance, the naturally existing comparable examples are mainly collected from coding problems, while the generated examples can belong to other sources, such as a large software project. In addition, the programs in the naturally existing example are self-contained, while both the Java and Python programs in the generated example contain user-defined classes and external functions that are defined elsewhere, including "HeaderWriter", "HeaderWriter.write()", and "align()". Besides, programs in one naturally existing comparable example typically have the same output format (e.g., "YES" or "NO" in this case), while programs we generate

| | Java-to-Python | | | Python-to-Java | | |
|---|---|---|---|---|---|---|
| | **BLEU** | **CB** | **CA@1** | **BLEU** | **CB** | **CA@1** |
| **Best baseline** | 79.9 | 73.2 | 69.4 | 81.4 | 78.4 | 62.0 |
| **CMTrans** | 82.1 ± 0.4** | 76.1 ± 1.4** | 72.1 ± 1.6** | 84.3 ± 0.4** | 82.0 ± 0.5** | 64.6 ± 1.0** |

| | C++-to-Python | | | Python-to-C++ | | |
|---|---|---|---|---|---|---|
| | **BLEU** | **CB** | **CA@1** | **BLEU** | **CB** | **CA@1** |
| **Best baseline** | 82.9 | 75.4 | 62.5 | 79.8 | 77.9 | 59.1 |
| **CMTrans** | 83.8 ± 0.6** | 76.8 ± 0.3** | 64.6 ± 0.8** | 82.1 ± 0.9** | 79.8 ± 0.5** | 63.2 ± 1.4** |

| | Java-to-C++ | | | C++-to-Java | | |
|---|---|---|---|---|---|---|
| | **BLEU** | **CB** | **CA@1** | **BLEU** | **CB** | **CA@1** |
| **Best baseline** | 89.3 | 88.5 | 84.6 | 90.9 | 90.1 | 68.3 |
| **CMTrans** | 89.9 ± 0.4* | 88.1 ± 0.2 | 83.7 ± 1.1 | 91.1 ± 0.4 | 90.3 ± 0.1** | 69.9 ± 1.1* |

Table 7: T-test results. We copy the results from the papers of the best baselines and run CMTrans for 3 different random seeds. **means significant results with p-value < 0.01. *means significant results with p-value < 0.05.

from the documentation may have different output formats, especially when the docstring is unclear about the return value. With all the differences, the combination of naturally existing and generated comparable corpora cover programs with a large variety of styles and domains, which provides diverse training signals to our model.

### A.4 Case Studies for Model-generated References

Figure 11 shows an example of the multiple references generated by CMTrans. The example is from the AVATAR-para dataset. We use our model to generate 50 candidate translations, 23 of which are correct (i.e., have exactly the same output as the source program on all the automatically generated test cases). We show the 9 correct translations with the highest probabilities, 4 of which are selected to the reference set.

**Quality of the Generated References**. In this example, the source program contains a for loop and an if statement. Note that the for loop has "$\leq$ high" as the end condition. When we translate it to the "for $ITER_VAR in range()" grammar in Python, we have to instead use "high + 1" as the end value. This is correctly handled in several different candidate translations.

**Diversity of the Generated References**. There are two main options for translating a loop: using either a for loop or a while loop. In the example in Figure 11, the selected reference set contains 2 translations with a while loop and 3 with a for loop,

including the ground truth translation.

We can observe that the differences between two candidates can be as small as adding or deleting parentheses and semicolons. For instance, the only difference between Candidate #3 and Candidate #9 is the parentheses around "i%10 == k". In this example, by computing the string edit distance, we avoid having both Candidate #3 and #9 in the reference set.

```
Obviously, the best way to buy food for every pet is to buy maximum possible food
for dogs and cats, then max(0, x-a) dogs and max(0, y-b) cats will not get food.
We will buy universal food for these dogs and cats. Then the answer is YES, if
max(0,x-a)+max(0,y-b)<=c, and NO else.
                                                <NL Instruction of the Coding Problem>
```

```cpp
#include <iostream>
#include<cstdio>
using namespace std;
int main(){
  int a, b, c, x, y, t;
  cin >> t;
  while ( t -- ){
    cin >> a >> b >> c >> x >> y;
    if ( a + c >= x ){
      if ( a >= x ) c = c;
      else c -= x - a;
      if (b + c >= y) cout << "YES\n";
      else cout << "NO\n";
    }
    else cout << "NO\n";
  }
}
                                <C++ Solution (User A)>
```

```java
import java.util.*;
public class Main{
 public static void main(String[] args){
    Scanner sc = new Scanner(System.in);

    for (int t = sc.nextInt(); t-->0;)
    {
      int a=sc.nextInt(),
      b=sc.nextInt(), c=sc.nextInt(),
      x=sc.nextInt(), y=sc.nextInt();
      int A = Math.max(0, x-a);
      int B = Math.max(0, y-b);
      System.out.println(
        c >= A + B ? "YES" : "NO");
    }
  }
}
                                <Java Solution (User B)>
```

```python
for _ in range(int(input())):
  a, b, c, x, y =
    map(int, input().split())
  if a >= x and b >= y:
     print( 'YES' )
  elif a >= x and b < y:
     if b + c < y:
        print( 'NO' )
     else:
        print( 'YES' )
  elif a < x and b >= y:
     if a + c < x:
        print( 'NO' )
     else:
        print( 'YES' )
  else:
     v = c - ((x-a) + (y-b))
     if v >= 0 :
        print( 'YES' )
     else :
        print( 'NO' )
                                <Python Solution (User C)>
```

Figure 9: A comparable example collected from a coding problem. We highlight the lines in the user-written C++, Java, and Python programs that can be matched.

```
Claim length of the term buffer for writing in the message with zero copy semantics.
@Parameters:
header - for writing the default header.
length - of the message to be written.
bufferClaim - to be updated with the claimed region.
activeTermId - used for flow control.
@Returns: the resulting offset of the term after the append on success otherwise FAILED.
                                                                <Input NL Documentation>
```

```java
public int claim(final HeaderWriter header, final int length,
      final BufferClaim bufferClaim, final int activeTermId){
 final int frameLength = length + HEADER_LENGTH;
 final int alignedLength = align(frameLength, FRAME_ALIGNMENT);
 final UnsafeBuffer termBuffer = this.termBuffer;
 final int termLength = termBuffer.capacity ();
 final long rawTail = getAndAddRawTail(alignedLength);
 final int termId = termId(rawTail);
 final long termOffset = rawTail & 0xFFFF_FFFFL;
 checkTerm ( activeTermId, termId );
 long resultingOffset = termOffset + alignedLength;
 if ( resultingOffset > termLength ){
   resultingOffset = handleEndOfLogCondition(termBuffer,
                  termOffset, header, termLength, termId);
 }
 else {
  final int frameOffset = (int)termOffset;
  header.write(termBuffer, frameOffset, frameLength, termId);
  bufferClaim.wrap(termBuffer, frameOffset, frameLength);
 }
 return (int)resultingOffset;
}
                                                <Input Java>
```

```python
def claim(header, length, bufferClaim, activeTermId):
  frameLength = length + HEADER_LENGTH
  alignedLength = align(frameLength, FRAME_ALIGNMENT)
  termBuffer = bufferClaim.termBuffer
  termLength = termBuffer.capacity()
  termId = termBuffer.active[activeTermId]
  termOffset = termBuffer.active[activeTermId] & 0xFFFF_FFFFL
  termOffset = handleEndOfLogCondition(termBuffer, termOffset,
                        header, termLength, termId);
  return termOffset
def handleEndOfLogCondition(termBuffer, termOffset,
                      header,termLength,termId):
  if termOffset + alignedLength > termLength:
     return termLength
  else :
     termBuffer.active[ activeTermId ] += 1
     termBuffer.active[ activeTermId ] %= termLength
     frameLength = termLength + FRAME_ALIGNMENT
     header.write(termBuffer, termOffset, frameLength, termId)
     bufferClaim.wrap( termBuffer, termOffset, frameLength)
     termBuffer.active[ activeTermId ] = 0
     termOffset += FRAME_ALIGNMENT
     return termOffset
                                                <Generated Python>
```

Figure 10: A comparable example generated by our method. The natural language documentation is the docstring of the Java program. We highlight the lines in the Java program and the generated Python program that can be matched.

| Input Source program (Java) |
|---|

```java
public static int counLastDigitK(int low, int high, int k){
  int count = 0;
  for ( int i = low; i <= high; i ++ )
    if ( i%10 == k ) count ++;
  return count ;
}
```

| Ground Truth Reference Translation (Python) |
|---|

```python
def counLastDigitK(low, high, k):
    count = 0
    for i in range( low, high+1 ):
        if ( i%10 == k ) :
            count += 1
    return count
```

| Correct Candidate # 1 |
|---|

```python
def countLastDigitK(low, high, k):
    count = 0
    for i in range(low, high+1 ):
        if ( i%10 == k ):
            count += 1
    return count
```

| Correct Candidate # 2 ✅ |
|---|

```python
def counLastDigitK(low, high, k):
    count = 0
    for i in range(low, high+1):
        if i%10 == k:
            count = count + 1
    return count
```

| Correct Candidate # 3 |
|---|

```python
def counLastDigitK(low, high, k):
    count = 0
    for i in range(low, high+1, 1):
        if i%10 == k:
            count += 1
    return count
```

| Correct Candidate # 4 |
|---|

```python
def counLastDigitK(low, high, k):
    count = 0
    i = low
    while i <= high :
        if ( i%10 == k ):
            count += 1
        i = i + 1
    return count
```

| Correct Candidate # 5 ✅ |
|---|

```python
def counLastDigitK(low, high, k):
    count = 0
    i = low
    while ( i <= high ):
        if ( i%10 == k ):
            count += 1
        i = i + 1
    return count
```

| Correct Candidate # 6 ✅ |
|---|

```python
def counLastDigitK(low, high, k):
    count = 0;
    i = low;
    while i <= high:
        if i%10 == k:
            count += 1;
        i += 1;
    return count;
```

| Correct Candidate # 7 |
|---|

```python
def counLastDigitK(low, high, k):
    count = 0
    i = low
    while ( i <= high ):
        if ( i%10 == k ):
            count += 1
        i += 1
    return count
```

| Correct Candidate # 8 |
|---|

```python
def counLastDigitK(low, high, k):
    count = 0;
    i = low;
    while ( i <= high ):
        if ( i%10 == k ):
            count += 1;
        i += 1;
    return count;
```

| Correct Candidate # 9 ✅ |
|---|

```python
def counLastDigitK(low, high, k):
    count = 0
    for i in range(low, high+1, 1):
        if ( i%10 == k ):
            count += 1
    return count
```

Figure 11: An example of the multiple references we generate. We show the top 9 correct translations generated by our model with the highest probabilities. Candidates with a checkmark are those we select to add to the reference set. We add these translations one by one based on their string edit distance with all the existing translations.