# OpenReview forum: "Data Augmentation for Code Translation with Comparable Corpora and Multiple References"
_EMNLP/2023/Conference — EMNLP 2023 Findings_

### Official Review · Reviewer_PYWi · 2023-08-04

**Soundness:** 4

**Excitement:**

4: Strong: This paper deepens the understanding of some phenomenon or lowers the barriers to an existing research direction.

**Paper Topic And Main Contributions:**

The paper deals with code translation and suggests two methods of data augmentation in order to solve the problem of data scarcity. For this, they create comparable corpora by using a code generator for the target language, prompted based on a docstring. Secondly, they generate multiple references (as training material too) by using a fine-tuned model. The automatically generated references are filtered by using test cases.

This seems to be the first paper that uses comparable corpora for code translation. Both methods seem to have positive impact as compared to the baseline model and are additionally useful if the conditions are even more low resource.

**Questions For The Authors:**

Could you provide some statistical significance testing (e.g. bootstrap resampling) for the BLEU scores - at least for the ones that have less than one point difference?

**Reasons To Accept:**

* well written paper, good structure and easy to understand
* good experimental method and positive results
* thorough specification of hypotheses and analysis based on them

**Reasons To Reject:**

* substance: the authors do not suggest a new model but pipeline several models as a preprocessing step
* fit for the conference: I have a doubt because code translation is not really NLP but I have seen many similar papers in these venues, so I don’t think this is irrelevant to the conference
* this paper extends on prior experiments of a particular mode. For readers who are not very fit with those experiments, there are some gaps in the understanding.
* the authors train a model similar to a previous paper. No details of this model are given here, that might be an issue for reproducibility

**Reproducibility:**

3: Could reproduce the results with some difficulty. The settings of parameters are underspecified or subjectively determined; the training/evaluation data are not widely available.

**Reviewer Confidence:**

4: Quite sure. I tried to check the important points carefully. It's unlikely, though conceivable, that I missed something that should affect my ratings.

**Typos Grammar Style And Presentation Improvements:**

* figure 2 must be missing some lines, particularly the coloured ones.
* the construct overlap statistics are not very well described
* line 450, the number 15 must be a typo, the figure shows 150

---

> ### Author Rebuttal · Authors · 2023-08-29
>
> Thank you for your thoughtful review of our paper! We address your concerns and answer your questions as follows:
>
> &nbsp;
> ### New pipeline, not new model
> The contribution of our paper is not just presenting a pipeline that reaches state-of-the-art results, but **the study and analysis of data augmentation methods of code generation**. Our two proposed data augmentation methods — training on comparable corpora and with multiple references — can be applied to a wide range of code generation models. Our analyses including what the model learns from comparable corpora and multiple references also provide insights into what the model learns in code translation training.
>
> &nbsp;
> ### Fit for the conference
> We agree that other papers on code translation that were published in NLP venues, such as [1][2], help demonstrate the fit of our paper. There was also a “NLP for programming” workshop co-located with ACL 2021 (NLP4Prog 2021), which focuses on deep learning based techniques for modeling both natural language and computer programs. We categorize code translation as a subdomain of “NLP for programming”.  Additionally, our paper draws motivation from classic methods in machine translation (the use of comparable corpora).
>
> [1] Wasi Uddin Ahmad, Saikat Chakraborty, Baishakhi Ray, and Kai-Wei Chang. 2022. Summarize and generate to back-translate: Unsupervised translation of programming languages. In Conference of the European Chapter of the Association for Computational Linguistics.
>
> [2] Wasi Uddin Ahmad, Md Golam Rahman Tushar, Saikat Chakraborty, Kai-Wei Chang. 2022. AVATAR: A Parallel Corpus for Java-Python Program Translation. In Findings of the Association for Computational Linguistics: ACL 2023.
>
> &nbsp;
> ### Writing and Reproducibility
> In the next version of the paper, we will revise the parts that cause confusion for you and other reviewers, such as experimental details and descriptions of our method.
>
> We will release the code and data upon acceptance.
>
> &nbsp;
> ### Clarification of Figure 2 (Questions for the authors, point 1)
> Could you be more specific about what Figure 2 is missing? We highlight the lines with similar functionalities in the same color. Take lines with the green background as an example, the input Python code contains a line: “```if "BB" in x or "WW" in x:```”. Its parallel example (in Java) has a similar line “```if (x.contains(“BB") || x.contains("WW")){```”, which has exactly the same logic and algorithm. Similarly, the comparable example (in Java) has a line “```if ( s.charAt(j) == s.charAt(j+1) ){```”, which also checks whether the board has contiguous “B” or “W”, but with a different algorithm.
> We will add line numbers in the next version to make it clearer.
>
> &nbsp;
> ### Significance Test (Questions for the authors)
> We present the **t-test** results as follows, where we run each experiment with 3 random seeds when finetuning on parallel data:
>
> &nbsp;
> ###
> **Java-to-Python:**
> |               |     BLEU     |      CB      |     CA@1     |
> |---------------|--------------|--------------|--------------|
> | Best baseline |     79.9     |     73.2     |     69.4     |
> | CMTrans       | 82.1 ± 1.4** | 76.1 ± 1.4** | 72.1 ± 1.6** |
>
> &nbsp;
> ###
> **Python-to-Java:**
> |               |     BLEU     |      CB      |     CA@1     |
> |---------------|--------------|--------------|--------------|
> | Best baseline |     81.4     |     78.4     |     62.0     |
> | CMTrans       | 84.3 ± 0.4** | 82.0 ± 0.5** | 64.6 ± 1.0** |
>
> &nbsp;
> ###
> **C++-to-Java:**
> |               |    BLEU    |      CB      |     CA@1    |
> |---------------|------------|--------------|-------------|
> | Best baseline |    90.9    |     90.1     |     68.3    |
> | CMTrans       | 91.1 ± 0.4 | 90.3 ± 0.1** | 69.9 ± 1.1* |
>
> &nbsp;
> ###
> **C++-to-Python:**
> |               |     BLEU     |      CB      |     CA@1     |
> |---------------|--------------|--------------|--------------|
> | Best baseline |     82.9     |     75.4     |     62.5     |
> | CMTrans       | 83.8 ± 0.6** | 76.8 ± 0.3** | 64.6 ± 0.8** |
>
> &nbsp;
> ###
> **Java-to-C++**
> |               |     BLEU    |     CB     |    CA@1    |
> |---------------|-------------|------------|------------|
> | Best baseline |     89.3    |    88.5    |    84.6    |
> | CMTrans       | 89.9 ± 0.4* | 88.1 ± 0.2 | 83.7 ± 1.1 |
>
> &nbsp;
> ###
> **Python-to-C++**
> |               |     BLEU     |      CB      |     CA@1     |
> |---------------|--------------|--------------|--------------|
> | Best baseline |     79.8     |     77.9     |     59.1     |
> | CMTrans       | 82.1 ± 0.9** | 79.8 ± 0.5** | 63.2 ± 1.4** |
>
> **means significant results with p-value < 0.01
>
> *means significant results with p-value < 0.05
>
> &nbsp;
> ###
>
> Note that we use computational accuracy (CA) as the primary evaluation metric because it directly reflects the functional correctness of the program, while BLEU and CodeBLEU (CB) only compare overlap with references. Our model significantly outperforms the baselines on CA@1 under 5 out of 6 language pairs.
>
> &nbsp;
> ### The construct overlap statistics (Questions for the authors, point 2)
> We compute the Construct Overlap Rate for each type of construct as follows: for each test example, we consider the ground truth label as 1 if the source input contains this type of construct, and 0 otherwise. Similarly, we consider the prediction result as 1 if the translation output by some model contains this type of construct and 0 otherwise. Then we compute the accuracy, recall, and F1 scores for each type of construct. The “construct overlap rate” in Figure 6 is the recall score.
>
> We will also report the construct overlap accuracy and F1 in the next version of the paper. Below are the accuracy, recall, and F1 scores for CMTrans (CC. Only) (no finetuning).
>
> &nbsp;
> ###
> **Java-to-Python:**
> |                                                   |  loop  |   if   | else if |
> |---------------------------------------------------|:------:|:------:|:-------:|
> | Accuracy                                          | 0.9582 | 0.9119 |  0.2857 |
> | Recall  (the Construct Overlap Rate in our paper) | 0.8418 | 0.7461 |  0.2162 |
> | F1                                                | 0.8962 | 0.8207 |  0.2462 |
>
> &nbsp;
> ###
> **Python-to-Java:**
> |                                                   |  loop  |   if   | else if |
> |---------------------------------------------------|:------:|:------:|:-------:|
> | Accuracy                                          | 0.8744 | 0.8822 |  0.4444 |
> | Recall  (the Construct Overlap Rate in our paper) | 0.9972 | 0.7868 |  0.1081 |
> | F1                                                | 0.9318 | 0.8317 |  0.1739 |
>
>
> We will revise the Figure captions and the explanations of the experiments in the next version.
>
> &nbsp;
> ### Typo in line 450 (Questions for the authors, point 3)
> Line 450 does not contain “15”. Do you mean the “our model generates more than 15 unique translations for 214 test examples” in Line 459?
> We are sorry for the confusion. We intended to describe the statistics of the “16-20” group in Figure 7(a). In Java-to-Python translation, there are in total 464 test examples. For 214 of them, CMTrans (MultiRef. Only) generate >=16 unique translations. In contrast, CodeT5 and TransCoder-ST-ft only generate >=16 unique translations for 145 and 146 examples.
> We will rephrase the result descriptions here in the next version.

---

### Official Review · Reviewer_jFgc · 2023-08-07

**Soundness:** 4

**Excitement:**

3: Ambivalent: It has merits (e.g., it reports state-of-the-art results, the idea is nice), but there are key weaknesses (e.g., it describes incremental work), and it can significantly benefit from another round of revision. However, I won't object to accepting it if my co-reviewers champion it.

**Paper Topic And Main Contributions:**

The paper focuses on machine translation of code.
The goal is to translate (for example) Python code into Java code.
To do this, the authors use existing benchmarks. But the parallel data for training is not huge.
To solve this problem, they propose two ideas:

1) they generate comparable corpora, train a model on them, and tune it on an existing parallel corpus. For the generation of comparable corpora, they use the human user documentation of the source file, and they automatically generate code in the target language based on this documentation.

2) They enrich the corpus with several target references. To do this, they ask the system to automatically generate several translations. They select the translations that pass the automatically generated unit test. And they select translations in order to promote diversity.

The authors propose numerous experiments and study the results according to each of their proposals (separately), and they describe the behaviour of their system according to hyperparameters. The system is compared with several state-of-the-art systems.

**Questions For The Authors:**

jFgc_A. In fact, unlike natural language, formal grammars for programming languages are fully defined. So why not program a "compiler" from language A to language B, instead of using deep learning or machine learning?

jFgc_B. How could your proposition and previous state-of-the-art propositions work for very different language (for example, Python versus Prolog)?

jFgc_C. The parallel data is aligned line-by-line. But several lines are not aligned. Are this lines cut off before training?

jFgc_D. Could you discuss the quality of docstrings? How did you train the generative model (lines 228-231)? What is its performance?

jFgc_E.Table 2: give precision about significance of results

jFgc_F. Table 2: you copy/paste previous results. Are you sure that the condition tests are strictly the same?

jFgc_G. lines 389-392 : Comparable corpora allow to build fluent translations, not precise translations according to the input. Therefore, the effectiveness of models trained on comparable corpora should be lower. Contrary to intuition, the results are better than baseline models. The proposed explanation is unconvincing.

jFgc_H. Figure 6 : loop, if, else-if are present in the vast majority of program, I guess. Even by chance, a model should obtain more than 21%, shouldn't it? Please, could you comment this?

jFgc_I. lines 485-488: the explanation is not clear. Why could the training be unbalanced?

jFgc_J. Limitations: what is the bias given by competitive programming websites.

**Reasons To Accept:**

The paper is well written and easy to follow. It is not too technical.
The experiments are rich and try to explore several pertinent questions. Past results from 7 systems are given for comparison.
Several propositions are evaluated, and each one is separately evaluated, this allows to have more refined results.

**Reasons To Reject:**

The usefulness of deep learning and machine learning is not discussed. In fact, unlike natural language, formal grammars for programming languages are fully defined. So why not program a "compiler" from language A to language B?

The explanations of the results are not convincing:
1) Lines 389-392 : Comparable corpora allow to build fluent translations, not precise translations according to the input. Therefore, the effectiveness of models trained on comparable corpora should be lower. Contrary to intuition, the results are better than baseline models. The proposed explanation is unconvincing.

Edit: the given explanation is convincing. This should be added to the camera ready paper

2) The explanation of lines 485-488 is not clear to me.

Edit: the given explanation is now clear, and should be added to the camera ready paper

The results are sometimes very close to the baseline models, and no information is given about the significance.

Edit: the authors gave this information.

**Reproducibility:**

3: Could reproduce the results with some difficulty. The settings of parameters are underspecified or subjectively determined; the training/evaluation data are not widely available.

**Reviewer Confidence:**

3: Pretty sure, but there's a chance I missed something. Although I have a good feel for this area in general, I did not carefully check the paper's details, e.g., the math, experimental design, or novelty.

---

> ### Author Rebuttal · Authors · 2023-08-29
>
> Thank you for your thoughtful and insightful feedback on our paper. We address your concerns and answer your questions as follows:
>
> &nbsp;
> ### Using compilers for code translation (Reason to reject, point 1, and also jFgc_A)
> It is reasonable to build a compiler for code translation. However, such transcompilers are time-consuming to develop, requiring expertise in both source and target languages. Transcompilers rely on manual rewrite rules applied to the abstract syntax tree (AST) of the source code. Furthermore, building transcompilers is especially hard for language pairs with different amounts of information or levels of abstraction. An example is Python-to-Java translation, where variable type information is required for Java when declaring variables, but is not required and usually omitted in Python.
>
> &nbsp;
> ### Explanation of Results (Reason to reject, point 2, and also jFgc_E, jFgc_G, and jFgc_I)
> *(jFgc_G)* For **lines 389-392**, just a reminder, our final model is trained first on comparable corpora and then on parallel data. While we agree that training on just the comparable corpora would lead to unsatisfactory performance, we show that models trained on comparable corpora followed by parallel data **perform better and converge faster** than models trained on just parallel data. Our analysis of the fluency of output (Line 423-435) indicates that the reason may be that training on comparable corpora already increases the fluency of the translation, so the model will not need so much of the limited parallel data to improve both the fluency and the accuracy of the translation.
>
> *(jFgc_I)* For **lines 485-488**, while generating multiple references we initially generate the same number of candidates for each program, but then we **filter out incorrect translations, which vary in number across programs**, leading to an unequal number of references per program in the filtered data.
>
> *(jFgc_E)* We present the **t-test** results as follows, where we run each experiment with 3 random seeds when finetuning on parallel data:
>
> &nbsp;
> ###
> **Java-to-Python:**
> |               |     BLEU     |      CB      |     CA@1     |
> |---------------|--------------|--------------|--------------|
> | Best baseline |     79.9     |     73.2     |     69.4     |
> | CMTrans       | 82.1 ± 1.4** | 76.1 ± 1.4** | 72.1 ± 1.6** |
>
> &nbsp;
> ###
> **Python-to-Java:**
> |               |     BLEU     |      CB      |     CA@1     |
> |---------------|--------------|--------------|--------------|
> | Best baseline |     81.4     |     78.4     |     62.0     |
> | CMTrans       | 84.3 ± 0.4** | 82.0 ± 0.5** | 64.6 ± 1.0** |
>
> &nbsp;
> ###
> **C++-to-Java:**
> |               |    BLEU    |      CB      |     CA@1    |
> |---------------|------------|--------------|-------------|
> | Best baseline |    90.9    |     90.1     |     68.3    |
> | CMTrans       | 91.1 ± 0.4 | 90.3 ± 0.1** | 69.9 ± 1.1* |
>
> &nbsp;
> ###
> **C++-to-Python:**
> |               |     BLEU     |      CB      |     CA@1     |
> |---------------|--------------|--------------|--------------|
> | Best baseline |     82.9     |     75.4     |     62.5     |
> | CMTrans       | 83.8 ± 0.6** | 76.8 ± 0.3** | 64.6 ± 0.8** |
>
> &nbsp;
> ###
> **Java-to-C++**
> |               |     BLEU    |     CB     |    CA@1    |
> |---------------|-------------|------------|------------|
> | Best baseline |     89.3    |    88.5    |    84.6    |
> | CMTrans       | 89.9 ± 0.4* | 88.1 ± 0.2 | 83.7 ± 1.1 |
>
> &nbsp;
> ###
> **Python-to-C++**
> |               |     BLEU     |      CB      |     CA@1     |
> |---------------|--------------|--------------|--------------|
> | Best baseline |     79.8     |     77.9     |     59.1     |
> | CMTrans       | 82.1 ± 0.9** | 79.8 ± 0.5** | 63.2 ± 1.4** |
>
>
>
> **means significant results with p-value < 0.01
>
> *means significant results with p-value < 0.05
>
> &nbsp;
> ###
> Note that we use computational accuracy (CA) as the primary evaluation metric because it directly reflects the functional correctness of the program, while BLEU and CodeBLEU (CB) only compare text overlap with references. Our model significantly outperforms the baselines on BLEU, CodeBLEU, and CA@1 for most language pairs, and our improvements are typically large on CA@1.
>
>
> &nbsp;
> ### Questions for the authors
>
> Translations between very different languages (jFgc_B): Our work and the majority of previous code translation works train a Seq2Seq model to translate between two languages. The details of training are provided in Lines 241-254. In principle, we can train a Seq2Seq model to translate other language pairs as well. An example of “very different languages” is the translation between programming languages and natural languages, which are called code summarization and code generation. Previous work [1] shows that Seq2Seq models trained for code summarization and code generation can also have strong results.
>
> Unaligned lines in parallel data (jFgc_C): We do not cut off any lines. The reason is removing such lines leads to influent and “unnatural” code pieces in the truncated data. Training with such code will cause inconsistency with testing, where the input will typically be fluent and complete code.
>
> Quality of docstrings (jFgc_D): We use the CodeSearchNet dataset for the functions with docstrings [2]. They apply preprocessing such as removing short docstrings and truncating long ones to improve the quality of docstrings.
>
> We do not train the code generation model by ourselves but apply CodeRL finetuned on MBPP (for translations involving Python) and CodeT5 finetuned on CONCODE (for other translations) as the code generation models. More details can be found in the Appendix (Lines 736-749).
>
> Test cases used by baseline models (jFgc_F): Assuming that condition tests refer to the test cases, yes – we use the same test cases as all baseline methods, which are provided by the TransCoder dataset.
>
> Comparison to random chance in construct overlap analysis (jFgc_H): We compute the percentage of each type of construct in the test set:
>
> **Java-to-Python:**
> |                                                              |  loop  |   if   | Else if |
> |--------------------------------------------------------------|:------:|:------:|:-------:|
> | Percent of test examples with this construct                 | 76.29% | 68.75% |  7.97%  |
> | Construct overlap rate of CMTrans (CC. Only) (no finetuning) | 84.18% | 74.61% |  21.62% |
> | Construct overlap rate of CodeT5 (no finetuning)             | 28.53% | 35.42% |    0    |
>
> **Python-to-Java:**
> |                                                              |  loop  |   if   | Else if |
> |--------------------------------------------------------------|:------:|:------:|:-------:|
> | Percent of test examples with this construct                 | 75.31% | 69.08% |  7.68%  |
> | Construct overlap rate of CMTrans (CC. Only) (no finetuning) | 99.72% | 78.68% |  10.81% |
> | Construct overlap rate of CodeT5 (no finetuning)             | 14.60% | 36.64% |    0    |
>
>
> All the numbers are lower than the construct overlap rate of CMTrans (CC. Only) (no finetuning) and higher than that of CodeT5 (no finetuning).
> As shown in the experiment in Figure 5, when the input and output programming languages are different, CodeT5 (no finetuning) is not able to generate fluent code in the target language. As a result, even if the percentage of loops or if structures are high in the training data, without finetuning, CodeT5 may not learn to generate the code with the same distribution.
>
>
> Bias of programming websites (jFgc_J): The code on competitive programming websites mostly uses only standard inbuilt libraries. They are also typically short in length and do not import functions from external files.
> All the above properties typically do not hold for other code pieces on GitHub. However, as these websites provide functional tests, they have become a standard benchmark for evaluating code translation, where functional tests are crucial to evaluate accuracy of the translated functions.
>
> Reproducibility: We will release the code and data upon acceptance.
>
>
> ### References
> [1] Shuai Lu, Daya Guo, Shuo Ren, Junjie Huang, Alexey Svyatkovskiy, Ambrosio Blanco, Colin Clement, Dawn Drain, Daxin Jiang, Duyu Tang, Ge Li, Lidong Zhou, Linjun Shou, Long Zhou, Michele Tufano, Ming Gong, Ming Zhou, Nan Duan, Neel Sundaresan, Shao Kun Deng, Shengyu Fu, Shujie Liu. CodeXGLUE: A Machine Learning Benchmark Dataset for Code Understanding and Generation. Arxiv. abs/2102.04664.
>
> [2] Hamel Husain, Ho-Hsiang Wu, Tiferet Gazit, Miltiadis Allamanis, and Marc Brockschmidt. 2020. Codesearchnet challenge: Evaluating the state of semantic code search.

---

### Official Review · Reviewer_9cbG · 2023-08-11

**Soundness:** 3

**Excitement:**

2: Mediocre: This paper makes marginal contributions (vs non-contemporaneous work), so I would rather not see it in the conference.

**Paper Topic And Main Contributions:**

This paper focuses on program translation tasks and tries to generate more parallel training data automatically for supervised translation training.  The authors use two methods to generate code pairs: 1) use docstring of one of program code method (program A) to generate another program (program B) to get A-B as supervised data.  2) based on existing parallel code, they use a model to convert source code to multiple candidates for a target program language, then select some of these candidates as multi-references codes.

**Questions For The Authors:**

Have you experimented with current large language models like ChatGPT and GPT-4 on this dataset? I'm intrigued to know how your work fares in comparison to these models.

**Reasons To Accept:**

1. Code translation tasks pose greater challenges compared to the more mature field of neural machine translation. As code-assisted software like Copilot gains prominence, the potential for future research in this area is immense.
2. The earlier work, Transcoder-ST, introduced an effective approach for code translation tasks by automatically generating code pairs, such as back translation, to create an extensive range of potential data pairs for the task. Subsequently, filtering was performed using compilation tools like unit tests to overcome the issue of scarce labeled data. The author further enhances this method by incorporating docstrings as a supervisory signal, resulting in improved performance.

**Reasons To Reject:**

1. Limited novelty: One of the challenges for code translation is the scarce parallel training data. This paper addresses the issue by using models to generate parallel code. Specifically, it employs docstrings (summarized text) as a bridge for source-target code alignment to generate related parallel code for model training. However, this idea has already been explored in the work "Summarize and Generate to Back-translate: Unsupervised Translation of Programming Languages."
2. Limited scalability: Utilizing docstrings for code generation restricts scalability, as it requires that source code functions contain docstrings. Furthermore, the multi-reference code generation relies on test cases to select the correct candidates, making it difficult to apply this method to labeled datasets without test cases.
3. GPT potential: Considering the effectiveness of GPT3.5/GPT4 in code generation, Why not use GPT as parallel code generator to create parallel pairs?
4. Experimental design: This work conducts all experiments based on the CodeT5 model, using only a small evaluation dataset (TransCoder dataset). In the original CodeT5 paper, the model is evaluated on the CodeXGLUE benchmark for C#-Java translation without testing on the TransCoder dataset. Conversely, the TransCoder model is pretrained on C++/Java/Python and evaluated on their dataset. It’s strange to select CodeT5 as base model to validate the proposed methods on TransCoder dataset.

**Reproducibility:**

3: Could reproduce the results with some difficulty. The settings of parameters are underspecified or subjectively determined; the training/evaluation data are not widely available.

**Reviewer Confidence:**

5: Positive that my evaluation is correct. I read the paper very carefully and I am very familiar with related work.

---

> ### Author Rebuttal · Authors · 2023-08-29
>
> Many thanks for your thoughtful comments and valuable suggestions!
>
> * Here are the key points of our response:
>     * [Clarification] We emphasize the difference between our work and Transcoder-ST
>     * [Response to “Limited novelty”] We highlight the novelty of our first contribution compared to the “Summarize and Generate” EACL’23 paper
>     * [Response to “Limited scalability”] We clarify the details of our method
>     * [Response to “GPT potential”] We discuss why we do not use GPT models in this work
>     * [Response to “Experimental design”] We discuss the selection of base model and dataset
>
> &nbsp;
> ### Clarification: Difference to TransCoder-ST
> * We’d like to further highlight the difference between our work and Transcoder-ST. The main contributions of our work are two-fold:
>     * (1) Training on comparable corpora before finetuning (there are multiple means of creating comparable corpora, where leveraging docstrings is only one of them);
>     * (2) Augmenting the finetuning set with multiple references.
>
> TransCoder-ST uses neither the comparable corpora nor multiple references.
>
> &nbsp;
> ### Response to “Limited novelty”
> Thanks for pointing out the connection between our first contribution and the “Summarize and Generate” (S&G) EACL’23 paper. We agree that S&G can also be viewed as one way to create comparable corpora. Nevertheless, **the contribution of our work lies in conducting a more in-depth study of when and how to use comparable corpora for code translation**.
>
> “When and how to use comparable corpora” is a broad topic in natural language translation. After [1] introduced the definition and potential utility of comparable corpora, there have been research works that identify scenarios where comparable corpora do not improve the performance and study the conditions when they are beneficial [2][3]. There are also a series of works that present different ways to collect comparable corpora [4][5].
>
>
> For the study of **when** to use comparable corpora, **we present a surprising result that training on comparable corpora is beneficial even if there is already high-quality parallel data** (i.e., the finetuning set). This goes beyond the conclusion of the S&G paper that comparable corpora obtained by summarization and generation are beneficial when no other parallel data is available.
>
> **We also provide an analysis on why comparable corpora are beneficial**. Specifically, Sec 4.4 shows that by training on comparable corpora, the pretrained model learns to (1) produce fluent code in the target language, and (2) produce code with matching constructs as the source input (e.g., if statements, for loop, etc). The S&G paper does not provide analysis on what can be learned from the comparable corpora.
>
>
> For the study of **how** to use comparable corpora, we show that **code translation can benefit from multiple types of comparable corpora, and construction from docstring is only one of them**. We additionally study a type of naturally existing comparable corpora: different submissions to the same online coding problem. Experimental results show that both types of comparable corpora can improve performance, and combining them gives the best results.
>
> During the response period, we further explore two other types of comparable corpora: (1) KNN: retrieve k code pieces in the target language that have the most similar sequence embeddings to the source code, and (2) Random: a random piece of code in the target language. Experiments below show that both of them can improve the performance of Java-Python translation:
>
> **Java-to-Python**:
>
> | CC      | Implementation Details                                                            | Finetuning data | CA (Pass@1, k=10) |
> |---------|-----------------------------------------------------------------------------------|-----------------|-------------------|
> | —-      |                                                                                   | AVATAR-para     | 0.6099            |
> | Random  | Each Java function in AVATAR is paired up with a random Python function in AVATAR | AVATAR-para     | 0.6466            |
> | KNN     | Each Java function in AVATAR is paired up with K nearest python functions         | AVATAR-para     | 0.6530            |
>
>
> **Python-to-Java**:
>
> | CC      | Implementation Details                                                            | Finetuning data | CA (Pass@1, k=10) |
> |---------|-----------------------------------------------------------------------------------|-----------------|-------------------|
> | —-      |                                                                                   | AVATAR-para     | 0.5203            |
> | Random  | Each Python function in AVATAR is paired up with a random Java function in AVATAR | AVATAR-para     | 0.5456            |
> | KNN     | Each Python function in AVATAR is paired up with K nearest Java functions         | AVATAR-para     | 0.5809            |
>
>
> Our analysis in Sec. 4.4 helps explain why “Random comparable corpora” is also beneficial: the model learns to generate fluent code in the target language when the input is in the source language. In other words, the model learns to capture the statistical patterns and relationships between tokens in the target language.
>
> We will include this analysis in the next version of the paper.
>
> &nbsp;
> ### Response to “Limited scalability”
> Regarding the concern about the scalability of using docstrings to construct comparable corpora, we want to note that one can use any programs publicly available from GitHub that have docstrings for this purpose, which is already of a large scale. In our experiments, we only used 22K functions from the CodeSearchNet dataset. Large-scale GitHub datasets are publicly available, e.g. The Stack [6]
>
> Regarding multi-reference code generation, the test cases are **automatically** generated using fuzzing, so our method can be applied to any dataset of functions to obtain multiple references. The process of automatically generating test cases is described in Line 284-295. We also provide an illustration in “step 2: test cases generation” in the right half of Figure 3.
>
> &nbsp;
> ### Response to “GPT potential”
> There are two reasons why we do not use GPT models in our training. First, it’s expensive to apply/query GPT models at our scale. More importantly, **we don’t know whether GPT has already been trained on the test data we use**. Our test set, the TransCoder dataset, is collected from the GeeksforGeeks website. A large number of GeeksforGeeks code pieces have been crawled and uploaded to GitHub, which GPT-3.5 (and presumably GPT-4) are trained on. Our work uses only fully open-access models with known training data.
>
> &nbsp;
> ### Response to “Experimental design”
> For the choice of base model, we have tried TransCoder in early experiments and found that the performance of fine-tuned CodeT5 is slightly better than fine-tuned TransCoder.
>
> For the evaluation dataset, we do not use CodeXGLUE because it does not provide test cases and only supports CodeBLEU/BLEU as evaluation metrics. These metrics are less accurate than execution-based metrics such as Computational Accuracy. For example, semantically equivalent programs with different implementations will have low CodeBLEU and BLEU scores. In our experiments (Table 2), CodeT5 has higher BLEU and CodeBLEU scores than TransCoder-ST in Java-to-Python translation (79.4/72.5 vs. 73.1/68.7 BLEU/CodeBLEU), but much lower Computational Accuracy (61.0 vs. 68.5 CA@1).
>
> For the above reason, most code translation papers (TransCoder, DOBF, TransCoder-ST, S&G, AVATAR) use only TransCoder as the test dataset and we are following their setting.
>
> &nbsp;
> ### References
> [1] Dragos Stefan Munteanu and Daniel Marcu. 2005. Improving machine translation performance by exploiting non-parallel corpora. Computational Linguistics.
>
> [2] T. Etchegoyhen and Harritxu Gete. 2020. Handle with care: A case study in comparable corpora exploitation for neural machine translation. In International Conference on Language Resources and Evaluation.
>
> [3] Harritxu Gete and Thierry Etchegoyhen. 2022. Making the most of comparable corpora in neural machine translation: a case study. Language Resources and Evaluation, 56(3):943–971.
>
> [4] Monica Lestari Paramita, David Guthrie, E. Kanoulas, Robert J. Gaizauskas, Paul D. Clough, and Mark Sanderson. 2013. Methods for collection and evaluation of comparable documents. In Building and Using Comparable Corpora
>
> [5] Krzysztof Wołk, Emilia Rejmund, and Krzysztof Marasek. 2015. Harvesting comparable corpora and
> mining them for equivalent bilingual sentences using statistical classification and analogy-based heuristics. ArXiv, abs/1511.06285.
>
> [6] Denis Kocetkov, Raymond Li, Loubna Ben Allal, Jia Li, Chenghao Mou, Carlos Muñoz Ferrandis, Yacine Jernite, Margaret Mitchell, Sean Hughes, Thomas Wolf, Dzmitry Bahdanau, Leandro von Werra, Harm de Vries, The Stack: 3 TB of permissively licensed source code. ArXiv, abs/2211.15533.

---

### Meta-Review · Area_Chair_Gau8 · 2023-09-21

**Recommendation:** 3

**Metareview:**

This paper presents data augmentation techniques to generate comparable corpora for using machine translation for code translation (e.g. from Python into Java). The paper is well written, and the experimental results show that their approach outperforms CodeT5.

As the paper is heavily based on previous work, the contributions of the present work and its novelty, as compared to the previous studies, should be made more  clear.  The confusion regarding the experimental design should be clarified. Statistical significance testing should be added to the paper (thank you for providing the numbers in the author response).

---

### Decision · Program_Chairs · 2023-10-07

**Decision:**

Accept-Findings

**Comment:**

This paper presents data augmentation techniques to generate comparable corpora for using machine translation for code translation (e.g. from Python into Java). The paper is well written, and the experimental results show that their approach outperforms CodeT5.

As the paper is heavily based on previous work, the contributions of the present work and its novelty, as compared to the previous studies, should be made more  clear.  The confusion regarding the experimental design should be clarified. Statistical significance testing should be added to the paper (thank you for providing the numbers in the author response).